# MEPs elicited by multidirectional rotational-field TMS show marked differences compared to unidirectional Figure-of-8 and H7 coils

Orit Wonderman Bar Sela[1,2☯*], Shay Ofir Geva[2,3☯], Gaby S. Pell[4,5], Yiftach Roth[4,5], Jason Friedman[6], Afnan Muhana[2,7], Silvi Frenkel-Toledo[1,2‡], Nachum Soroker[2,3‡]

1 Department of Physical Therapy, School of Health Sciences, Ariel University, Ariel, Israel, 2 Brain-Rehab Lab, Department of Neurological Rehabilitation, Loewenstein Rehabilitation Medical Center, Raanana, Israel, 3 Gray Faculty of Medical and Health Sciences, Tel-Aviv University, Tel-Aviv, Israel, 4 BrainsWay Ltd., Jerusalem, Israel, 5 Department of Life Sciences, Ben Gurion University of the Negev, Beer Sheva, Israel, 6 Department of Physical Therapy, School of Health Professions, Gray Faculty of Medical and Health Sciences, Tel-Aviv University, Tel-Aviv, Israel, 7 Department of Occupational Therapy, School of Health Professions, Gray Faculty of Medical and Health Sciences, Tel-Aviv University, Tel-Aviv, Israel

☯ These authors are contributed equally to this work.
‡ SFT and NS contributed equally to this work.
* oritwonder1@gmail.com

## Abstract

Unidirectional transcranial magnetic stimulation (udTMS; e.g., via Figure-of-8 coil) depolarizes mainly neurons whose axonal orientation aligns with the direction of the induced electric field. A novel dual H-coil (T360°) TMS system (BrainsWayTM, Israel) generates a rotational magnetic field aimed to recruit a larger neuronal population by induction of a multidirectional electric field (rfTMS). This study aimed to comparatively assess the neurophysiological properties of motor evoked potentials (MEPs) elicited from the first dorsal interosseous (FDI) muscle following udTMS (via Figure-of-8 and H7 coils) vs. multidirectional rfTMS. In this study, 10 healthy adult subjects received TMS via the three coil configurations in a random order. The results showed that rfTMS elicited larger MEPs at a lower resting motor threshold (rMT) compared to the unidirectional coils. These findings suggest that rfTMS is likely to recruit larger populations of neurons compared to conventional udTMS coil configurations. This may be advantageous in efforts to enhance motor recovery following brain damage by treatments using TMS.

## Introduction

Transcranial magnetic stimulation (TMS) is a widely used diagnostic tool in clinical neurophysiology [1], a therapeutic intervention in different brain disorders [2,3], and a promising modality for enhancement of adaptive neuroplasticity and motor recovery post stroke [4,5]. TMS recruits mainly neurons whose axons run in parallel to the electric field induced in the brain. When delivered via conventional Figure-of-8 coils,

**Data availability statement:** All data are fully available without restriction. The minimal anonymized dataset necessary to replicate the study findings is publicly available at https://github.com/ShayOfir/MEP_IOcurve.

**Funding:** Loewenstein Rehabilitation Medical Center KM 600010312 Dr Nachum Soroker Ariel University KM 600010385 Ms Silvi Frenkel-Toledo.

**Competing interests:** I have read the journal's policy and the authors of this manuscript have the following competing interests:Authors YR and GP are employed by BrainsWay Ltd. All other authors declare no conflict of interest associated with this work. Specifically, no financial or personal relationships with other people or organizations could inappropriately influence or bias this study.

the electric current is unidirectional and at close-to-threshold levels depolarization skips adjacent neurons with non-parallel orientation [6–8]. Hence, in unidirectional TMS (udTMS), activation is limited to a relatively small neuronal population within the stimulated area. The neural response at a given orientation of the coil will become more extensive at higher stimulation intensities, at the risk of more adverse side effects [7,8].

Unlike frequently used udTMS coils (e.g., the Figure-of-8 coil), which are limited to stimulating areas up to about 1 cm beneath the inner surface of the skull, udTMS coils of the H configuration can reach a depth of up to 4 cm [9], contingent on the specific H-coil in use; therefore, they are categorized as deep-TMS coils. Depth of stimulation is achieved by the coils' winding design, which extends in multiple planes within the helmet. Additionally, H-coils deliver stimulation to a larger area, potentially activating more neurons, whereas Figure-of-8 coils are focal at the same stimulator output [10].

Rotational-field TMS (rfTMS) uses two orthogonal H-coils with a 90-degree phase shift to create a rotating electric field. In contrast to the focal effect of standard udTMS, rfTMS is circularly polarized and rotates over an almost complete cycle. This circular polarization affects a broader cortical range, activating neurons in various orientations, with deeper penetration into the brain matter [6,11,12]. In the present study, a multi-directional rotational-field dual H-coil configuration (also termed TMS360°), a specific implementation of rfTMS, was utilized.

Damage to the corticospinal tract (CST) is the most important determinant of upper-limb function post stroke, with severe damage predicting low likelihood of late recovery. Evaluation of MEPs is widely used to assess the residual capacity for neural transmission in the CST. MEPs are therefore commonly used as key biomarkers to predict stroke recovery and to stratify patients for personalized rehabilitation approaches [13,14], particularly in the context of interventions incorporating non-invasive brain stimulation [4]. However, MEP properties are influenced not only by CST integrity but also by stimulation-related factors (e.g., inter-trial interval, stimulation intensity, coil orientation relative to the hotspot, the spatial spread of the electric field induced by TMS), and by individual differences in neuroanatomy and motor-system responsiveness. Additionally, MEP amplitudes show noticeable trial-to-trial variability when recorded at a single stimulation intensity. Thus, previous researchers have proposed that assessing MEPs across a range of stimulus intensities using input–output (I/O) curves can elucidate cortical excitability more than single-intensity measurements [15–17]. The sigmoidal I/O relationship measures recruitment gain, maximum corticospinal output, and the dynamic range of excitability at different intensities. It may also mitigate sensitivity to temporary fluctuations in brain state and strengthen test-retest reliability.

With conventional unidirectional Figure-of-8 TMS, neuronal recruitment is highly sensitive to the orientation of the induced electric field relative to cortical geometry, potentially limiting activation of corticospinal neurons in orientations that differ from that of the induced electric field [8]. Multidirectional or rotational-field stimulation may reduce this orientation dependence by capturing spatially distinct neuronal

populations, thereby increasing the likelihood and robustness of measurable MEPs [11]. In this context, understanding how coil configuration influences MEP properties is critical to interpreting MEP-based biomarkers in stroke rehabilitation.

TMS has been experimentally applied in stroke rehabilitation for many years, with the aim of enhancing adaptive neuroplasticity and improving functional outcomes. In the majority of cases, stimulation is delivered via the Figure-of-8 coil. Response variability is large, and randomized control studies yield mixed results [4]. Enhanced robustness of neuroplastic effects through deeper and wider neuronal recruitment may pose a significant advantage for TMS-based therapies, yet exacerbation of side effects and the potential activation of adjacent inhibitory networks is a matter of concern [12]. Selection of the optimal stimulation target for an individual hemiparetic stroke patient is, to date, an unsolved issue. Clinical algorithms, like the widely used PREP2 [18] recommend stimulating the lesioned/non-lesioned hemisphere on the basis of the status of motor evoked potentials (MEP+/MEP-, respectively) plus other clinical biomarkers. Yet, whether this binary approach truly optimizes prognostication and clinical decision making was recently questioned [14], and alternative approaches were suggested [19]. Given the importance of MEP status in clinical rehabilitation practice, the extent to which MEP properties depend on TMS coil configuration was investigated. Specifically, properties of MEPs elicited from the contralateral first-dorsal interosseous (FDI) muscle were assessed. A comparison was made between udTMS coils (Figure-of-8 and H7) and the multi-directional dual H rotational field TMS coil (rfTMS, also termed T360°).

## Methods

### Participants

Ten healthy right-handed volunteers (7 males), with an age range of 30−73 (Mean±SD: 54.5±14.6) were recruited after screening for putative risks (e.g., history of epilepsy; presence of metal fragments). The study was approved by the Helsinki Committee of the Loewenstein Rehabilitation Medical Center (LRMC), IRB protocol number 0022-21-LOE. Participants were recruited from September 1, 2024, to February 1, 2025, comprising the entire duration of data collection for the study. Written informed consent was obtained from each participant and signed records were kept on file.

### Procedure

A crossover design was used, applying single-pulse TMS (spTMS) over the left- and right-hand regions of the primary motor cortex. This was carried out via three different coil configurations (Figure-of-8, H7, T360°) in random order (total number of conditions 2x3, separated by 15–30 minutes), spread over one or two testing sessions (see Fig 1A, 1B). Prior to data collection, a series of random numbers was generated using a simple randomization procedure in Microsoft Excel to determine the order of coil and hemisphere conditions. This sequence was manually reviewed to promote a balanced distribution of conditions across participants.

Hotspot localization and resting motor threshold (rMT) determination were conducted separately for each coil type and for each tested hemisphere. During this process, a systematic positional search was performed by incrementally moving the coil in anterior–posterior (1 cm) and medial–lateral (0.5 cm) directions while maintaining a constant orientation. Hotspot localization was performed independently for the Figure-of-8 coil and the BrainsWay coils. At each tested position, at least 2–3 single TMS pulses were delivered before determining whether the location elicited a reliable MEP. The motor hotspot was defined as the scalp location that consistently produced the largest and most stable MEP amplitudes at a fixed stimulator output. Neuro-navigation was not used; instead, standardized anatomical landmarks and cap-based grid coordinates were applied.

Resting motor threshold (rMT) was determined using an adaptive threshold-hunting procedure implemented via the Motor Threshold Assessment Tool (MTAT 2.0; Awiszus & Borckardt), [20] rather than the conventional '5-out-of-10' method. The initial stimulator intensity was set based on the output used during hotspot localization. The threshold was defined according to the predefined MEP amplitude criterion (100 μV), as detailed above, in accordance with the adaptive

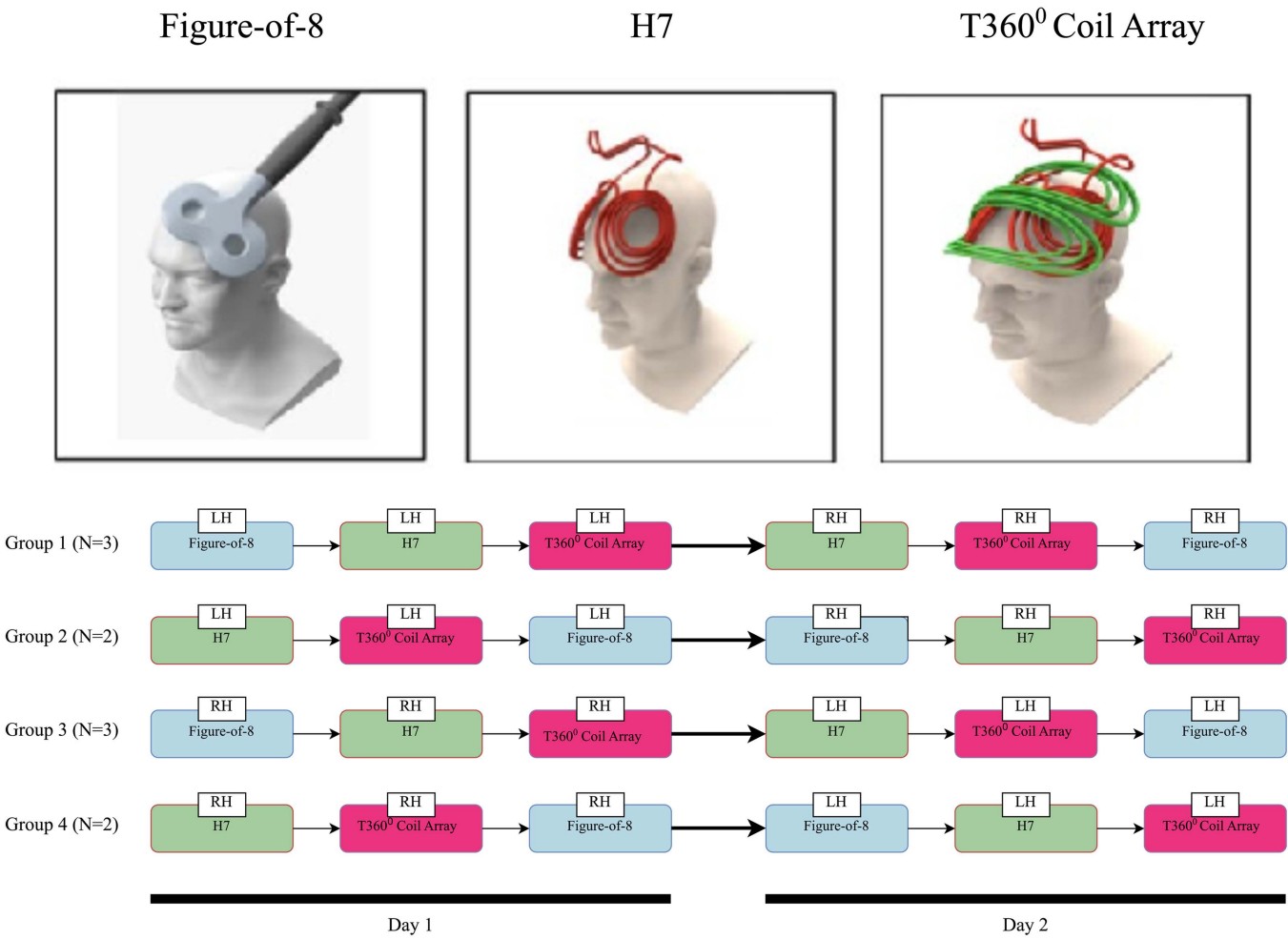

**Fig 1. (A) Coil geometry.** Schematic illustration of the unidirectional (Figure-of-8 and H7) and multidirectional rotational-field (T360°) TMS coil configurations used in the study. Reprinted by permission from Roth et al. (2023). **(B) Study design and crossover experimental setup.** Three TMS coils were tested in a randomized, crossover design with counterbalanced order. The rMT and MEP amplitudes were measured for each coil. These tests were conducted on 10 subjects across two sessions, each on a different day.

algorithm's convergence rules. This method has been shown to provide accurate and reliable rMT estimates while reducing the total number of TMS pulses required.

The order of coil configurations and the hemisphere side were counterbalanced across participants and were determined using simple random number generation. In addition to the single TMS pulses required for determining the hot spot and the rMT, five stimuli at seven stimulation intensities were delivered to the FDI hotspot. Stimulation intensities were fixed and ranged from 90% to 150% of the specific rMT for the condition (subject, coil and side) in 10% increments. All stimulations were delivered with a fixed inter-stimulus interval (ISI) between 5 and 7 seconds [21,22].

Overall, the study employed a randomized cross-over design, such that all participants were exposed to all coil configurations while serving as their own control. This counterbalancing procedure was implemented to minimize carryover effects and promote randomness across coil type and hemisphere. Fig 1 provides a schematic illustration of coil geometry (Fig 1A) and experimental workflow (Fig 1B).

## Transcranial magnetic stimulation (TMS)

Figure-of-8 coil (70 mm) was used with a Magstim SuperRapid2 stimulator (Magstim Ltd., Whitland, UK). The H7 and the T360° coil configurations were used with the BrainsWay Deep TMS System Model 108 (BrainsWay Ltd., Jerusalem, Israel) (see Fig 1A). The 'hotspot' was determined by standard anatomical landmarks and responsiveness to stimulation [20].

## Neurophysiological measures

The following measures were obtained: (a) resting motor threshold (rMT) and (b) peak-to-peak MEP amplitude [calculated as in [20,23]. MEPs were recorded from the resting FDI muscle upon stimulation of the contralateral primary motor cortex (M1). Ag/AgCl surface electrodes were placed in a belly-to-tendon montage (Delsys™ Inc., Boston, USA). EMG response triggered by spTMS was captured within a standard time window (from −500 to +500 ms) from trigger onset, with a 2KHz sampling rate, after amplification and bandpass filtering (1–65 Hz), using a Power-Lab C device and the LabChart software package (AD Instruments Ltd., New Zealand). MEP occurrence was confirmed when a post-trigger peak-to-peak EMG signal was greater than 0.1mV [22] and was valid morphologically on inspection and without excess noise. The onset of a MEP was defined as the time point where a rectified EMG signal exceeded two standard deviations of the rectified average EMG level recorded during the pre-stimulation interval [−100, 0 ms] [24]. All pre processing was performed by custom MATLAB scripts (The MathWorks, Natick, MA, USA; version R2024a). For each coil configuration (Figure-of-8, H7, T360°) and stimulation side (hemisphere – left, right), we computed the rMT using the algorithm from the TMS Motor Threshold Assessment Tool ver. 2.0 [20] and transformed it from % of the maximal stimulator output (MSO) to an 'adjusted' rMT value expressed in volts, using the formula:

$$rMT_{adjusted}(coil) = \frac{rMT \cdot V_{max}(coil)}{100}$$

where $V_{max}(coil)$ is the MSO in volts of the corresponding stimulator. In this study, all reported rMT values correspond to an adjusted rMT, determined by applying the above formula. The ratio of rMT values obtained with the T360° coil configuration was obtained relative to those acquired with the Figure-of-8 and H7 coils, using the formula:

$$Ratio = \frac{median\,(rMT_{T360°})}{median\,(rMT_{reference})}$$

where the reference was either the Figure-of-8 or the H7 coil.

## Statistical analysis

All analyses were performed using custom R scripts [25]. Assessment of the effect of coil and stimulation side on the rMT was done by a non-parametric two-way repeated-measures ANOVA (two within-subject factors) implemented in 'MANOVA.RM' package [26]. Post-hoc effects were evaluated using a permutations-based Fisher-Pittman one-way test [27] implemented in the 'coin' package. Input-Output (IO) curve analysis was done by modelling the effect of coil and side on the peak-to-peak amplitude across stimulator intensities (normalized to maximal stimulator output of the H7 and T360⁰ coil configurations to allow for comparison with the Figure-of-8 coil), using a Bayesian hierarchical model [28] of the following logistic function:

$$Y = \frac{A_{i,j,k}}{1 + e^{-s_{i,j,k}\left(I-\theta_{i,j,k}\right)}}$$

where: $Y$ corresponds to peak-to-peak amplitude, $A$ is the asymptotic MEP amplitude, $s$ is the slope, $I$ is the adjusted intensity level, and $\theta$ is the threshold curve. The indices $i$, $j$, and $k$ correspond to different subjects, coils, and sides, respectively. Fixed effects for coil type and stimulation side were included on all three parameters of the sigmoidal model ($A$, $\theta$, and $s$) to capture systematic condition-dependent differences in response amplitude, threshold, and slope. subject-level random effects were added to each parameter to account for inter-subject variability while retaining partial pooling across individuals. An interaction term between coil type and side was intentionally omitted from the hierarchical model, as incorporating this additional layer of complexity would have increased the risk of over-fitting given the available sample size and the already high dimensionality of the hierarchical model. The full details of the model are described in the supplementary material, including prior predictive, posterior predictive, convergence checks, and simulation-based checks. Sampling from the posterior distribution was done using Hamiltonian Monte-Carlo as implemented in Stan language and 'cmdstanr' R package with custom R scripts [29]. Significance levels were assessed by the highest posterior density interval (HPDI) using samples drawn from the posterior density function.

## Results

### The effect of coil and stimulation side on resting motor threshold

Two-way repeated-measures ANOVA with two within-subject factors (coil, side) revealed a significant main effect on the rMT for coil ($F_{1.31} = 98.26$, $p < 0.001$). The main effect of stimulation side was not found ($F_1 = 2.48$, $p = 0.115$) (see Fig 2), though there was a significant coil × side interaction effect ($F_{1.81} = 3.2$, $p = 0.046$). The differences between coils were assessed separately for each stimulation side using post-hoc analysis (Fisher-Pittman permutations test). This confirmed that the rMT obtained with the T360° coil configuration was significantly lower compared to both the Figure-of-8 coil (left: $Z = 3.03$, $p = 0.002$; right: $Z = 3.01$, $p = 0.0023$) and the H7 coil (left: $Z = 2.82$; $p = 0.0013$; right: $Z = 2.73$, $p = 0.0025$) (see Table 1 and Fig 2; Individual subject data is provided in Supporting Information S1 Fig). In addition, the H7 coil was associated with a significantly lower rMT compared to the Figure-of-8 coil (left $Z = 2.81$, $p = 0.0017$; right: $Z = 2.98$, $p = 0.0018$). As shown in Table 1, the ratios of median rMT values for the T360° coil configuration relative to the Figure-of-8 and H7 coils were 0.58 and 0.82, respectively, for the left hemisphere, and 0.56 and 0.86, respectively, for the right hemisphere.

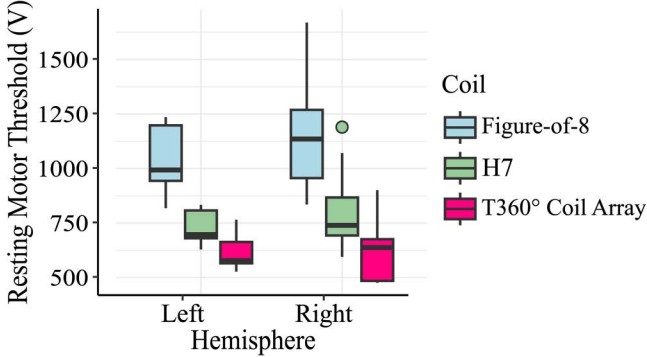

**Fig 2. Adjusted rMT according to hemisphere and coil type.** Adjusted rMT (volts) obtained by spTMS applied over the hand knob of the primary motor cortex with MEPs recruited from the contralateral FDI, comparing unidirectional (Figure-of-8 and H7) and multidirectional (rotational-field, T360°) coil configurations. The boxplots illustrate the median (line), first to third quartiles (Q1–Q3, box), whiskers extending up to 1.5 times the interquartile range (IQR), and dots represent outliers. The colors represent the coil configuration: Figure-of-8 (blue), H7 (green), and T360° (red).

**Table 1. Median (Q1, Q3) values of resting motor threshold (rMT) in volts (V) categorized by stimulated hemisphere and coil type.**

| Measure | Stimulated hemisphere | Coil | | |
|---|---|---|---|---|
| | | Figure-of-8 | H7 | T360° |
| rMT (V) | Left | 994 (944, 1198) | 697 (680, 808) | 578 (565, 636) |
| | Right | 1136 (956, 1269) | 740 (693, 867) | 638 (484, 676) |

### The effect of coil and side on the MEP input-output curve

We modeled the motor evoked potential (MEP) input–output (I/O) relationship using a hierarchical Bayesian logistic formulation, in which the three canonical parameters of the I/O curve—$\theta$ (threshold), $s$ (slope), and $A$ (asymptotic MEP amplitude)—were estimated for each coil type and side. In the context of TMS physiology, $\theta$ reflects the stimulation intensity at which corticospinal neurons begin to respond reliably, $s$ indexes the gain or recruitment efficiency of the corticospinal pathway, and $A$ represents the maximal attainable MEP amplitude under suprathreshold stimulation [20]. We applied 4 chains with 1,000 iterations for warm-up and an additional 1,000 iterations for sampling, with an adapt delta of 0.999 and a maximal tree depth of 20. All convergence diagnostics indicated that model converged well (R-hat < 1.01, ESS bulk > 400 and visual mixing was good according to trace plots [29]. (see please the linked Supporting Information). All results are reported in terms of posterior probability of contrasts between delta coefficients for the I/O curve parameters (median difference, credible interval, CI, and probability of direction of the contrast greater than zero, PD). The resultant I/O curves are depicted in Fig 3.

The Figure-of-8 coil showed higher thresholds ($\theta$) compared to both H7 (median difference = 0.71, 89% CI: 0.05–1.32; PD = 96%) and the T360° (median = 1.22, 89% CI: 0.57–1.95; PD = 99.72%). The H7–T360° contrast was more uncertain (median difference = 0.51, 89.1% CI: –0.19–1.15; PD = 89%).

Log-amplitude ($\log A$) value was lower for Figure-of-8 relative to both H7 (median difference = –0.77, 89% CI: –0.91 to –0.64; PD = 0%) and the T360° (median difference = –1.15, 89% CI: –1.30 to –0.99; PD = 0%). The H7–T360° contrast also indicated consistently lower values for H7 (median difference = –0.38, 89% CI: –0.54 to –0.21; PD = 0%).

With regard to the I/O curve slope ($s$), usage of the Figure-of-8 coil was associated with higher slope values than both H7 (median difference = 0.081, 89% CI: 0.062–0.102; PD = 100%) and the T360° (median difference = 0.115, 89% CI: 0.097–0.134; PD = 100%). The H7–T360° contrast also indicated consistently higher slopes for H7 (median difference = 0.034, 89% CI: 0.025–0.043; PD = 100%).

When comparing the posterior predictive distribution of left- and right-hemisphere stimulation, there was no consistent difference between the thresholds (median difference = −0.336, 89% CI: (−0.982) − 0.334; PD = 20.2%) and the slopes (median difference = 0.0047, 89% CI: (−0.0023) − 0.0131, PD = 83.5%). Nonetheless, left hemispheric stimulation was associated with a consistently greater log-amplitude (median difference = 0.594, 89% CI: 0.481–0.709; PD = 100%).

All subjects tolerated the study well, and there were no significant adverse events. However, physical reactions to TMS pulses were often more noticeable for the H7 and rfTMS coils compared with the Figure-8 coil, particularly in the form of blinking and facial twitching. This was expected given the wider spread of the electric field of deep TMS coils. One subject reported temporary tooth pain during the pulses, which quickly disappeared, and another subject reported a headache the day after.

### Discussion

Consistent with previous findings, single-pulse TMS delivered through a multidirectional rotational-field coil configuration (T360°) produced substantially lower rMT than the H7 coil [11]. Extending these findings, Bayesian modelling of

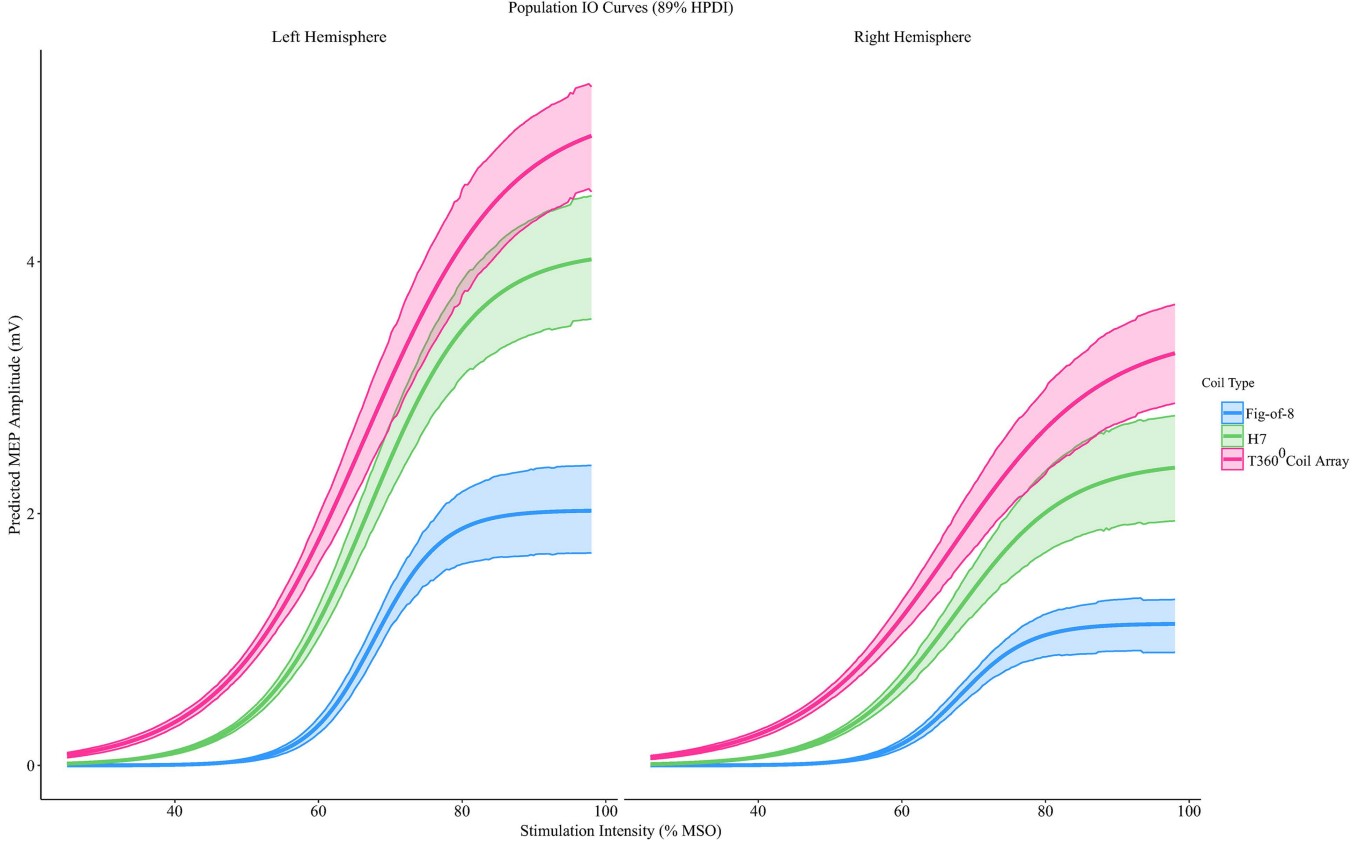

**Fig 3. Population-level input–output curves for three TMS coil types.** Population-level input–output (IO) curves depicting predicted motor evoked potential (MEP) amplitude (in mV) as a function of stimulation intensity (% of maximal stimulator output, MSO) for three TMS coil types: Figure-of-8 (blue), H7 (green), and T360° (red). Curves represent posterior predictions from a Bayesian hierarchical model incorporating fixed effects for coil configuration and hemisphere (left vs. right), and subject-level (10 healthy participants undergoing spTMS with MEPs recruited from the contralateral FDI muscle. Shaded regions indicate the 89% highest posterior density interval (HPDI).

the input–output (I/O) curve revealed a characteristic excitability profile marked by lower recruitment thresholds, larger asymptotic MEP amplitudes, and a more gradual recruitment slope for the T360° coil configuration relative to both H7 and Figure-of-8 coils. Together, these results demonstrate that the multidirectional rotational-field coil configuration elicits a distinct and robust pattern of corticospinal activation. The combination of lower thresholds, higher maximal MEP amplitudes, and a shallower I/O slope suggests that the rotational field coil configuration recruits a larger and more heterogeneously oriented neuronal population. Lower thresholds reflect earlier activation of axons aligned with the induced field, while higher maximal amplitudes indicate that more corticospinal neurons ultimately contribute to the response [17]. The shallower slope further implies gradual recruitment across neurons with diverse orientations and activation thresholds [15,16].

The broad, multidirectional stimulation of the T360° coil configuration may be advantageous for therapeutic neuromodulation, particularly in post-stroke populations in which networks are reorganized, and widespread activation enhances plasticity. With these advantages, rfTMS may indicate the existence of residual corticospinal neural transmission in cases of severe stroke where contralateral motor response cannot be elicited by unidirectional TMS. These advantages of rfTMS may also lead to improved outcomes in rehabilitation treatments employing rTMS, potentially decreasing the high response variability currently observed in such treatments, thus lowering the doubts about their

effectiveness [30]. Additionally, a lower rMT could lead to reduced patient discomfort during TMS sessions and mitigate the risk of adverse side effects associated with higher intensities of stimulation, which is a noted concern with elevated stimulation levels [7,8].

Unlike conventional TMS coils, which are limited to stimulating areas up to ~1 cm beneath the skull surface, deep TMS can reach a depth of up to ~4 cm, contingent on the specific H-coil in use [31]. While the Figure-of-8 coil produces a relatively superficial, focal electric field, the H-coil induces a deeper, broader field. Differences in the depth-focality of the coil types result in different volumes of tissue being stimulated with each coil [32]. In this study, the H7 coil yielded significantly lower rMT compared to the Figure-8 coil. The multidirectional rotational field T360° coil configuration is based on the H7 coil with an additional coil in orthogonal orientation, where the two coils are connected to separate stimulator channels and are operated with a phase delay, leading to a rotational field. As a result, the T360° coil configuration stimulates the same cortical volume [11] as the H7 coil but activates neurons at various orientations within that volume. This multi-orientation recruitment is reflected in the higher maximal MEP amplitudes and the more gradual I/O curve slope that was observed with the T360° coil configuration, indicating enhanced functional activation without deeper or broader stimulation.

A stimulation-side effect was observed, characterized by higher maximal MEP amplitudes in the left hemisphere, as indicated by the I/O curve. Hemispheric asymmetries in corticospinal excitability have been documented in previous research using various TMS paradigms and outcome measures. For example, steeper recruitment slopes have been reported in the non-dominant hemisphere in studies examining I/O curves [15], while larger excitatory motor representations have been suggested in the dominant hemisphere in some navigated TMS mapping studies [33,34]. Accordingly, these findings suggest that the effects of hemispheric differences on corticospinal output may vary across specific metrics and experimental approaches.

For diagnostic purposes, such as evaluating corticospinal tract integrity, establishing motor thresholds, or generating somatotopic MEP maps, a more focal and unidirectional stimulation is desirable. One option is the conventional Figure-of-8 coil due to its focality. Similarly, when operated at the threshold for motor activation, H-coils such as the H7 can produce focal stimulation at the motor cortex representation of specific movements, without activating additional movements. Moreover, by operation of the two orthogonal coils of the T360° coil configuration simultaneously (and not with a lag, as done to achieve a rotating field), and calibration of the relative amplitudes of the two channels, one can probe all the possible orientations without moving the coils. This may be a powerful tool for the investigation of brain networks and corticospinal tracts, both at rest and during a task, with comparable probing of all orientations. Yet, the possibility of stimulating areas beyond the intended target region cannot be ruled out, and given the broader spatial extent of activation by H and T360° coil configurations, adjacent cortical regions (e.g., premotor and supplementary motor cortices) might be activated. The functional and behavioral implications of such off-target activation remain poorly understood and warrant further investigation. Increases in MEPs should be interpreted cautiously and not taken as evidence of activation supporting improved motor function (i.e., superior corticospinal recruitment leading to enhanced treatment response) [35].

A recent study has indicated the potential benefits of repetitive TMS applied via the T360° coil configuration for patients with major depression [36]. Today, coils of the H type, either in single or dual (rfTMS) coil configuration, are in use mainly for psychiatric disorders. In contrast, the large majority of studies that have applied TMS to enhance motor recovery post stroke used the Figure-of-8 coil type [4]. However, significant response variability and uncertainty about long-term efficacy have led to recent doubts [30] and have stressed the need for novel approaches. The capacity of rfTMS to recruit larger neuronal populations in multiple orientations at the stimulation site, with lower stimulator outputs [6,12], may advance the field of therapeutic brain stimulation for enhancement of motor recovery. Following a stroke, reorganization of motor networks involves multidirectional axonal sprouting and new synapse formation [37]. The increased ability of rfTMS to stimulate neurons in various orientations may thus have a positive impact on the likelihood of benefit from TMS-based rehabilitation therapies.

While this study introduces novel options for using rfTMS, it also has several limitations. The most notable are the small sample size and the wide age range of participants. The broad age range was intentional, aiming to reflect the typical clinical population. Age-related variability was partially accommodated in the I/O curve modelling through the inclusion of subject-level random effects. Moreover, the use of two different stimulators may pose a potential limitation, though small difference in MSOs was adjusted by normalizing values. To minimize potential systematic bias from time-dependent effects, the order of experimental conditions was randomized. Although we aimed to counter-balance conditions, the final cohort suffered from minor imbalances (Fig 1B). In addition, due to technical reasons associated with the practical finding of the motor spot, stimulation with H7 coil was always followed by the T360° coil configuration. These imbalances could affect the generalizability of our results. The inter-stimulus interval (ISI) used in this study was of 5–7 seconds, as done in previous studies [21,22]. While this ISI was important for minimizing variance attributed to subjects' fatigue, it might have introduced a bias due to cumulative neural effects. Lastly, although neuronavigation was not used in the present study, the hotspot location was verified using EMG for all three coils in each participant. Hence, any residual spatial uncertainty was shared across conditions and was unlikely to bias the within-subject comparison between coils.

The current study represents the initial phase of an ongoing project, where in the near future, therapeutic rTMS will be applied via the rotational-field coil configuration, in stroke patients with severe hemiparesis and chronic non-functional upper-limb, for the first time. We hope the advantages demonstrated in this preliminary study will prove beneficial for patients.

## Conclusion

The study corroborates and expands upon prior research, pointing to an advantage of delivering TMS via a rotational-field multi-directional coil configuration over unidirectional coils in standard use. The study shows that single-pulse TMS delivered via the T360° coil configuration elicits higher maximal MEP amplitude relative to two unidirectional coils - Figure-of-8 and H7. Another advantage is the lower recruitment threshold (rMT) associated with the rotational-field multi-directional coil configuration, which enables repetitive stimulation for therapeutic purposes at lower stimulation intensities and thus lower risk of side effects. The application of rfTMS to human subjects is likely to mitigate the orientation dependence of standard unidirectional TMS, thereby establishing a foundation for a potential role in advancing motor rehabilitation following brain damage.

## Supporting information

**S1 Fig. Individual subjects' data according to coil configuration and stimulated hemisphere.** Adjusted resting motor threshold (rMT, volts) upon single-pulse stimulation targeting the hand knob in left and right primary motor cortices, with unidirectional (Figure-of-8 and H7) and multidirectional (rotational-field, T360°) coil configurations and motor evoked potentials (MEPs) derived from the contralateral first dorsal interosseous (FDI) muscle. Each point in the graph represents a single subject, and the black lines join the same subject across the different coils for the same hemisphere (see Results section).
(TIF)

**S2 Fig. Electric field simulations for the three TMS coil configurations.** Simulations of the electric field using SimNIBS (Thielscher et al., 2015) software for a coil positioned over the hand knob of the primary motor cortex (C3). The stimulator outputs are scaled according to the motor threshold (see Table 1). The smaller extent of stimulation of the Figure-of-8 coil relative to the H7 coil reflects differences in stimulation efficiency, given the reduced motor threshold in the latter. Note that SimNIBS is a "static" field simulation and cannot, therefore, simulate the rotational invariance of the T360° coil configuration.
(TIF)

## Acknowledgments

This work was carried out by the first author, OWBS, in partial fulfillment of the requirements for a PhD degree at Ariel University, Israel, under the supervision of authors SFT and NS. We thank Dr Corinne Serfaty, Head of the Neurological Department at the Loewenstein Rehabilitation Medical Center, for her most valuable support. We thank Prof Abraham Zangen from Ben-Gurion University of the Negev for his advice and support. We thank the ten friends and family members who volunteered to participate.

## Author contributions

**Conceptualization:** Orit Wonderman Bar-Sela, Gaby S. Pell, Yiftach Roth, Silvi Frenkel-Toledo, Nachum Soroker.

**Data curation:** Orit Wonderman Bar-Sela, Shay Ofir Geva, Afnan Muhana.

**Formal analysis:** Orit Wonderman Bar-Sela, Shay Ofir Geva, Gaby S. Pell, Yiftach Roth, Jason Friedman, Silvi Frenkel-Toledo, Nachum Soroker.

**Funding acquisition:** Silvi Frenkel-Toledo, Nachum Soroker.

**Methodology:** Orit Wonderman Bar-Sela, Shay Ofir Geva, Gaby S. Pell, Yiftach Roth, Jason Friedman, Afnan Muhana, Silvi Frenkel-Toledo, Nachum Soroker.

**Project administration:** Orit Wonderman Bar-Sela, Silvi Frenkel-Toledo, Nachum Soroker.

**Resources:** Gaby S. Pell, Yiftach Roth, Silvi Frenkel-Toledo, Nachum Soroker.

**Software:** Jason Friedman.

**Supervision:** Silvi Frenkel-Toledo, Nachum Soroker.

**Writing – original draft:** Orit Wonderman Bar-Sela, Nachum Soroker.

**Writing – review & editing:** Orit Wonderman Bar-Sela, Shay Ofir Geva, Gaby S. Pell, Yiftach Roth, Silvi Frenkel-Toledo, Nachum Soroker.

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
