## [Decision Letter · Decision Letter 0]

10 Dec 2025

Thank you for submitting your manuscript to PLOS ONE. After careful consideration, we feel that it has merit but does not fully meet PLOS ONE’s publication criteria as it currently stands. Therefore, we invite you to submit a revised version of the manuscript that addresses the points raised during the review process.

We look forward to receiving your revised manuscript.

Kind regards,

Mu-Hong Chen, M.D., Ph.D.

Academic Editor

PLOS One

Journal Requirements:

[Funding was received from the LRMC research foundation and from Ariel University to SFT and NS.].

4. Please expand the acronym “LRMC” (as indicated in your financial disclosure) so that it states the name of your funders in full.

[I have read the journal's policy and the authors of this manuscript have the following competing interests:Authors YR and GP are employed by BrainsWay Ltd. All other authors declare no conflict of interest associated with this work. Specifically, no financial or personal relationships with other people or organizations could inappropriately influence or bias this study.].

We note that one or more of the authors are employed by a commercial company: Brainsway Ltd.

6. We noted in your submission details that a portion of your manuscript may have been presented or published elsewhere. [It was uploaded as a preprint, DOI: 10.36227/techrxiv.175356536.62169355/v1] Please clarify whether this publication was peer-reviewed and formally published. If this work was previously peer-reviewed and published, in the cover letter please provide the reason that this work does not constitute dual publication and should be included in the current manuscript.

7. When completing the data availability statement of the submission form, you indicated that you will make your data available on acceptance. We strongly recommend all authors decide on a data sharing plan before acceptance, as the process can be lengthy and hold up publication timelines. Please note that, though access restrictions are acceptable now, your entire data will need to be made freely accessible if your manuscript is accepted for publication. This policy applies to all data except where public deposition would breach compliance with the protocol approved by your research ethics board. If you are unable to adhere to our open data policy, please kindly revise your statement to explain your reasoning and we will seek the editor's input on an exemption. Please be assured that, once you have provided your new statement, the assessment of your exemption will not hold up the peer review process.

8. We note that Figure 1 in your submission contains copyrighted images. All PLOS content is published under the Creative Commons Attribution License (CC BY 4.0), which means that the manuscript, images, and Supporting Information files will be freely available online, and any third party is permitted to access, download, copy, distribute, and use these materials in any way, even commercially, with proper attribution. For more information, see our copyright guidelines: http://journals.plos.org/plosone/s/licenses-and-copyright.

9. We notice that your supplementary figures are uploaded with the file type “Figure”. Please amend the file type to “Supporting Information”. Please ensure that each Supporting Information file has a legend listed in the manuscript after the references list.

Reviewers' comments:

Reviewer's Responses to Questions

**Comments to the Author**

1. Is the manuscript technically sound, and do the data support the conclusions?

Reviewer #1: Yes

Reviewer #2: Partly

Reviewer #3: Partly

2. Has the statistical analysis been performed appropriately and rigorously?

Reviewer #1: Yes

Reviewer #2: Yes

Reviewer #3: Yes

3. Have the authors made all data underlying the findings in their manuscript fully available?

Reviewer #1: Yes

Reviewer #2: Yes

Reviewer #3: Yes

4. Is the manuscript presented in an intelligible fashion and written in standard English?

Reviewer #1: Yes

Reviewer #2: Yes

Reviewer #3: Yes

Reviewer #1: This crossover study evaluated the neurophysiological characteristics of a newly developed rotational-field (RF) TMS coil by comparing motor evoked potentials (MEPs) across three coil configurations in 10 healthy adults. The RF coil yielded significantly lower resting motor thresholds and 3-6 fold higher MEP amplitudes relative to conventional figure-of-8 and H7 coils, suggesting enhanced neuronal recruitment through multidirectional stimulation. While these preliminary findings are encouraging for the development of RF coil technology, some suggestions are offered below to optimize the presentation and interpretation of the results.

- Method, Lines 97-99: ”random order (total number of conditions 2x3 separated by 15-30 minutes), spread over one or two testing sessions”

The authors state that the three coil configurations were tested "in a random order" but provide insufficient methodological detail. Please provide a complete description of the randomization procedure, such as the sequence generation method, whether block randomization is used to ensure balanced allocation, how the possible order is distributed among the 10 participants, and other details.

- Result, Lines 176-177: "Cases where stimulation intensities differed by more than 5% of the MSO were eliminated from analysis, thus yielding 281 EMG traces out of 1881 valid EMG traces"

The exclusion of 85% of valid EMG traces due to a 5% MSO difference threshold is problematic and unnecessarily reduces statistical power. The current method risks introducing selection bias, which could be more serious than the small intensity differences it attempts to control. Please include all 1881 valid traces using mixed-effects models with stimulation intensity as a covariate, or perform a sensitivity analysis using different exclusion thresholds (5%, 10%, 15% MSO) to demonstrate robustness

- Discussion, Lines 244-247: "We expect that this technique could provide beneficial and effective treatment by significantly boosting their potential for upper limb recovery"

The authors' strong prediction of treatment outcomes for stroke patients based solely on neurophysiological findings from only 10 healthy adults is clearly overly optimistic. In fact, no patient data are presented to support these clinical claims. The use of definitive language such as "significantly boosting" and "beneficial and effective treatment" is inappropriate for preliminary healthy subject data. Please revise "ALL" relevant content in the discussion section.

Reviewer #2: This study compares three different TMS coils (Figure-of-8, H7, and RF). The authors report that the RF coil yields a lower resting motor threshold and elicits larger motor evoked potentials compared to unidirectional coils. While the findings hold promise for future applications in stroke rehabilitation, the study is limited by a small sample size. I have the following questions and suggestions regarding this study.

1. The participants in this study span a broad age range (30-73 years). Given that age is known to significantly influence both resting motor threshold (rMT) and motor evoked potentials (MEPs), the current manuscript does not explain how this potential confounding factor was controlled for in the analysis. It is suggested that the authors add an explanation of how age was handled or explicitly discuss this in the Limitations section.

2. While the experimental design employs single-pulse TMS, specific parameters are missing, such as the inter-pulse interval. It is suggested that the authors provide these details. Additionally, the legend for Figure 1 labels the image as "Study design," yet the figure content only displays the shapes and setup of the different coils. Please verify whether the figure content or the legend requires modification to accurately reflect the content.

3. The interval between each condition was 15–30 minutes; is this sufficient to eliminate the effects of the previous stimulation? Furthermore, given the prolonged duration of the experiment, could this reduce the subjects' corticospinal excitability and introduce experimental biases?

4. Regarding TMS coil positioning, this study relied solely on traditional landmarks. In the absence of MRI-guided neuronavigation, is this sufficient to accurately target the primary motor (M1) region? Furthermore, how was it confirmed that the hot spot for the H7 and RF coils precisely corresponds to the M1 region?

5. In the Results section, the MEPs elicited by the H7 coil show significant asymmetry (0.187 mV for the right hemisphere, and 0.47 mV for the left) compared to the Figure-of-8 and RF coils. It is suggested that the authors explain the potential physiological or technical factors contributing to this asymmetry in the manuscript.

6. Additionally, the legend for Figure S1 mentions that 3 subjects were excluded from the right hemisphere analysis and 1 subject from the left hemisphere analysis; is this data exclusion related to the asymmetry in the H7 coil MEP results? It is recommended that the exclusion of these subjects and the reasons for doing so be explicitly mentioned in the manuscript.

Reviewer #3: The manuscript addressed an important methodological comparison of coil configurations and introduces a multidirectional rfTMS system. The results are promising, but several conceptual, methodological, and reporting issues must be clarified before publication.

Major comments:

1. The Introduction lacks a clearly articulated rationale connecting MEP properties, coil physics, and motor rehabilitation relevance. The authors should specify hypotheses regarding how multidirectional stimulation may improve recruitment or prognostic interpretation. For example, clearly define why MEPs matter for diagnosis/prognosis, explain how coil configuration might influence MEP detectability.

2. The procedural description requires greater specificity. For example, the manuscript does not state how many trials were delivered for each coil and hemisphere, nor does it clarify whether stimulations were randomized, blocked, or interleaved, or whether the order of coil configurations was counterbalanced across participants. The inter-trial interval (ITI) was also not reported. In addition, the description of hotspot determination was too brief; the authors should specify the method used to locate the hotspot (e.g., grid search procedure), the number of stimulations used to confirm it, whether neuronavigation was implemented, and whether the hotspot location differed across coil types. Similarly, the criteria used to define the resting motor threshold (rMT) were not described. It should be stated explicitly whether the standard definition (i.e., at least 5 out of 10 MEPs exceeding 50 µV) was used. Clarifying these points was essential for ensuring methodological transparency and replicability. In addition, the experimental procedure should be illustrated in figure 1.

3. Regarding safety and tolerability, the manuscript asserted that a lower rMT may reduce the risk of adverse effects, yet it does not report whether any side effects or discomfort actually occurred during the study. A brief section should be added describing whether participants experienced pain, muscle twitches outside the targeted FDI muscle, headaches, or any other sensations during stimulation. It would also be important to note whether the subjective experience of stimulation intensity differed across coil types. Including these details would strengthen the manuscript by providing a more complete account of participant safety and tolerability.

4. The Discussion acknowledged that H-coils stimulate deeper and broader cortical areas. However, the implications of this reduced focality for motor specificity remain insufficiently addressed. Existing electric-field models, including the simulations presented in Roth et al. (2020), indicated that rfTMS produced a substantially deeper and wider field compared with both figure-of-8 and single H-coils. Because the rotational field covered a large cortical volume and stimulated axons across multiple orientations, the authors should clarify how such broad activation aligned with the goal of targeting specific motor representations in M1.

A more detailed discussion was warranted on several conceptual points. First, how “focal” was the RF coil according to current modeling work, and could the multidirectional fields disrupt somatotopic specificity within the motor cortex? Second, given the coil’s broader spread, was there a realistic risk of activating adjacent or unintended regions such as the premotor cortex or supplementary motor area? Including or citing electric-field simulations would significantly strengthen the readers’ understanding of these issues.

Moreover, the authors’ interpretation of larger MEPs as indicative of greater corticospinal recruitment should be balanced with the recognition that rfTMS stimulated a larger tissue volume, which may amplify MEPs simply through increased spatial coverage rather than improved corticospinal integrity. In other words, a larger MEP does not necessarily translate to better function or superior treatment response. This distinction is important both scientifically and clinically.

5. Following the context drafted in comment 4, it may be valuable for the authors to differentiate between the potential uses of rfTMS in rehabilitation versus diagnostic assessment. The broad, multidirectional stimulation of the RF coil could be advantageous for therapeutic neuromodulation, particularly in post-stroke populations where networks are reorganized and where widespread activation may enhance plasticity or reveal residual pathways not captured by conventional TMS. However, for diagnostic purposes, such as evaluating corticospinal tract integrity, establishing motor thresholds, or generating somatotopic MEP maps, the conventional figure-of-8 coil might be superior due to its focality and interpretability. If the authors agree with this distinction, incorporating it into the Discussion or Limitations section would improve conceptual clarity and strengthen the manuscript’s translational relevance.

Minor comments:

1. The manuscript used first-person pronouns, which was not typical for academic writing. Replacing it with passive voice or a third-person description would be more formal.

2. The reference format was inconsistent. The mixed use of APA-like elements, URLs, and irregular punctuation made the reference list appear informal. A full revision of the references to meet journal standards is necessary.

3. The terminology used in the manuscript would be better to keep consistent, e.g., RF, T360o, and rfTMS, and ensure all abbreviations are defined upon first use.

**Do you want your identity to be public for this peer review?** For information about this choice, including consent withdrawal, please see our Privacy Policy

Reviewer #1: No

Reviewer #2: No

Reviewer #3: **Yes:** Yi-Chun Tsai

---

## [Author Response · Author response to Decision Letter 1]

30 Jan 2026

Please refer to the attached document titled “Response to Reviewers,” which includes a detailed, point-by-point reply to all comments made by the editor and reviewers.

---

## [Decision Letter · Decision Letter 1]

11 Feb 2026

MEPs elicited by multidirectional rotational-field TMS show marked differences compared to unidirectional figure-of-8 and H7 coils.

PONE-D-25-48772R1

Dear Dr. Orit Wonderman Bar-Sela,

We’re pleased to inform you that your manuscript has been judged scientifically suitable for publication and will be formally accepted for publication once it meets all outstanding technical requirements.

Kind regards,

Mu-Hong Chen, M.D., Ph.D.

Academic Editor

PLOS One

Additional Editor Comments (optional):

Reviewers' comments:

Reviewer's Responses to Questions

**Comments to the Author**

Reviewer #1: All comments have been addressed

Reviewer #2: All comments have been addressed

Reviewer #3: All comments have been addressed

2. Is the manuscript technically sound, and do the data support the conclusions?

Reviewer #1: Yes

Reviewer #2: Yes

Reviewer #3: Yes

3. Has the statistical analysis been performed appropriately and rigorously?

Reviewer #1: Yes

Reviewer #2: Yes

Reviewer #3: Yes

4. Have the authors made all data underlying the findings in their manuscript fully available?

Reviewer #1: Yes

Reviewer #2: Yes

Reviewer #3: Yes

5. Is the manuscript presented in an intelligible fashion and written in standard English?

Reviewer #1: Yes

Reviewer #2: Yes

Reviewer #3: Yes

Reviewer #1: The authors have adequately addressed the majority of my concerns in their revised manuscript. I have no further issues to raise.

Reviewer #2: The manuscript now presents a robust methodological framework. The authors have been transparent about the study's constraints. All points raised by this reviewer in the initial submission have been adequately considered and appropriately incorporated in the revised manuscript. This reviewer thanks the authors’ effort.

Reviewer #3: The revised Introduction is substantially improved in terms of linking MEP properties, coil physics and clinical relevance. However, the study aim and hypotheses remain somewhat implicit. I would recommend that the authors add 1–2 sentences at the end of the Introduction explicitly stating the primary research question and their directional hypotheses.

**Do you want your identity to be public for this peer review?** For information about this choice, including consent withdrawal, please see our Privacy Policy

Reviewer #1: No

Reviewer #2: No

Reviewer #3: **Yes:** Yi-Chun Tsai

---

## [Editor Report · Acceptance letter]

PONE-D-25-48772R1

PLOS One

Dear Dr. Wonderman Bar-Sela,

I'm pleased to inform you that your manuscript has been deemed suitable for publication in PLOS One. Congratulations! Your manuscript is now being handed over to our production team.

Kind regards,

on behalf of

Dr. Mu-Hong Chen

Academic Editor

PLOS One